# Nontuberculous Mycobacteria Prevalence in Aerosol and Spiders’ Webs in Karst Caves: Low Risk for Speleotherapy

**DOI:** 10.3390/microorganisms9122573

**Published:** 2021-12-13

**Authors:** Dana Hubelova, Vit Ulmann, Pavel Mikuska, Roman Licbinsky, Lukas Alexa, Helena Modra, Milan Gersl, Vladimir Babak, Ross Tim Weston, Ivo Pavlik

**Affiliations:** 1Faculty of Regional Development and International Studies, Mendel University in Brno, Tr. Generala Piky 7, 613 00 Brno, Czech Republic; dana.hubelova@mendelu.cz (D.H.); helena.modra@mendelu.cz (H.M.); 2Public Health Institute Ostrava, Partyzanske Nam. 7, 702 00 Ostrava, Czech Republic; vit.ulmann@zuova.cz; 3Institute of Analytical Chemistry of the CAS, Veveri 97, 602 00 Brno, Czech Republic; mikuska@iach.cz (P.M.); alexa@iach.cz (L.A.); 4Transport Research Centre, Lisenska 33a, 636 00 Brno, Czech Republic; roman.licbinsky@cdv.cz; 5Faculty of AgriSciences, Mendel University in Brno, Zemedelska 1/1665, 613 00 Brno, Czech Republic; milan.gersl@mendelu.cz; 6Veterinary Research Institute, Hudcova 70, 621 00 Brno, Czech Republic; babak@vri.cz; 7Department of Biochemistry and Genetics, La Trobe Institute for Molecular Science, La Trobe University, Bundoora, Melbourne, VIC 3086, Australia; R.Weston@latrobe.edu.au

**Keywords:** non-tuberculous mycobacteria (NTM), mycobacteria other than tuberculosis (MOTT), saprophytic environmental mycobacteria, risk groups of microorganisms, underground therapy, cave airflow, asthma therapy

## Abstract

A total of 152 aerosol and spider web samples were collected: 96 spider’s webs in karst areas in 4 European countries (Czech Republic, France, Italy, and Slovakia), specifically from the surface environment (*n* = 44), photic zones of caves (*n* = 26), and inside (aphotic zones) of caves (*n* = 26), 56 Particulate Matter (PM) samples from the *Sloupsko-Sosuvsky* Cave System (speleotherapy facility; *n* = 21) and from aerosol collected from the nearby city of *Brno* (*n* = 35) in the Czech Republic. Nontuberculous mycobacteria (NTM) were isolated from 13 (13.5%) spider’s webs: 5 isolates of saprophytic NTM (*Mycobacterium gordonae*, *M. kumamotonense*, *M. terrae*, and *M. terrae* complex) and 6 isolates of potentially pathogenic NTM (*M. avium* ssp. *hominissuis*, *M. fortuitum*, *M. intracellulare*, *M. peregrinum* and *M. triplex*). NTM were not isolated from PM collected from cave with the speleotherapy facility although mycobacterial DNA was detected in 8 (14.3%) samples. Temperature (8.2 °C, range 8.0–8.4 °C) and relative humidity (94.7%, range 93.6–96.6%) of air in this cave were relatively constant. The average PM_2.5_ and PM_10_ mass concentration was 5.49 µg m^−3^ and 11.1 µg m^−3^. Analysed anions (i.e., F^−^, Cl^−^, NO_2_^−^, SO_4_^2−^, PO_4_^3−^ and NO_3_^−^) originating largely from the burning of wood and coal for residential heating in nearby villages in the surrounding area. The air in the caves with speleotherapy facilities should be monitored with respect to NTM, PM and anions to ensure a safe environment.

## 1. Introduction

Speleotherapy is a climatic treatment method that uses the specific climate in the cave as a natural resource [1,2,3]. Air quality, underground climate, and radiation are considered the main therapeutic factors in speleotherapy in caves. The following components are relevant for a suitable cave climate used to treat asthma: absence of normal biotic conditions (e.g., light), temperature difference between the underground and surface environment, low air circulation (<0.1 m/s), presence of medium to high relative humidity air, high levels of air ionization, low dust levels (0.05 mg/m^3^), small amounts of pollen (<30 mg/m^3^), low levels of contamination by bacteria and other bioaerosols, climate stability, and the presence of the finest aerosols of vital elements (Na, K, Mg, Ca etc.) [4].

Speleotherapy is a treatment method that uses the specific properties of the natural environment of karst caves, or certain artificial underground spaces for treatment, i.e., to eliminate or alleviate functional disorders of certain systems of the human body. Current knowledge has shown that the complex of events in the cave environment has a positive effect on the human body through stimulation and modulation of the immune and autonomic nervous systems. Impaired balance between sympathetic and parasympathetic tone activity significantly affects the onset and progression of many diseases. The key healing properties of the cave microclimate include its constant temperature, humidity, small fluctuations in barometric pressure, clean air and high content of light ions [5].

The speleotherapy is form of alternative therapy is used for patients with bronchial asthma (*Asthma bronchiale*). This is a chronic respiratory disease whose development and course can be prevented. Bronchial asthma is referred to as a “complex disease” in which chronic inflammation of the bronchial wall is the underlying mechanism of the disease. The genetic background of about two-thirds of asthma sufferers contributes to the onset and clinical picture of asthma [4]. Bronchial asthma results from a combination of genetic susceptibility, environmental factors and exposure to specific allergens [6]. Speleotherapy can modify immunological parameters of treated patients. Changes in selected immunological parameters were studied and the impact of speleotherapy on cytokine-induced killer (CIK) cells was confirmed [7]. The positive benefit of speleotherapy for children with *asthma bronchiale* was demonstrated by Gaus and Weber in a study of 121 evaluated children aged 4–10 years [8].

Exposure to air pollution is known to have negative effects on human health including lung cancer, cardiovascular and respiratory diseases including bronchial asthma [9,10,11]. Atmospheric aerosol, defined as the suspensions of solid or liquid particles (Particulate Matter, PM) in an air, is regarded as the most important component of air pollution. Aerosol particles are composed of a complex mixture of different constituents such as inorganic species (i.e., elements, oxides, salts, ions etc.), organic compounds (i.e., PAHs, hopanes, organic acids, saccharides, alkanes, dioxins etc.), soot and biological components (pollen, fungi, viruses, bacteria, etc.). The composition of airborne particles in ambient air in populated areas is the subject of interest of various studies, however, there is only limited information on aerosol composition in specific microenvironments such as caves. Generally, the cave atmosphere is considered less polluted compared to the atmosphere of populated areas [12].

Patients with bronchial asthma require medical treatment. Available therapy is not always completely successful. Other complementary treatments are specific immunotherapy, respiratory rehabilitation, yoga, psychotherapy, speleotherapy, etc. [13]. The alternative speleotherapy has been proposed as a complementary treatment of patients with *asthma bronchiale* with benefits in treating the condition itself as well allowing the administration of medications to be reduced to a minimum [14,15,16]. The positive effect of speleotherapy usually manifested itself after 3–4 weeks of treatment with a daily stay in a cave lasting 3–6 h depending on the temperature in the underground environment [16]. The importance and positive results of speleotherapy were discussed in previous studies from the then Czechoslovakia, Poland and Hungary. Significant improvements in the health status of one-fifth of paediatric patients was reported [1].

The quality of the cave microclimate must comply with the strict requirements in the parameters issued by the Permanent Commission on Speleotherapy (PCS), which is part of the International Union of Speleology (UIS) [17,18]. The monitored parameters are a constant air temperature (7–8 °C) with minimum air flow, relatively high humidity (almost 100%), dust-free and sterility from microbial and allergenic particles, increased CO_2_ content, low pH, low concentration of negative ions, etc. Specifically monitoring the presence of mycobacteria is not included in these parameters.

Literature on speleotherapy in relation to mycobacterial infections in humans is limited. Only in the USA in the Mammoth Cave was experimental therapy performed on patients with human tuberculosis. In October 1842, John Croghan, M.D., believed that cave’s constant temperature and humidity might prove healthy for 15 patients suffering from pulmonary tuberculosis. In the spring of that year, patients were living in wooden and stone huts constructed along the main corridor of the cave. In 1843 the experiment was terminated. Several patients died in the cave and many others became sicker. Dr. John Croghan himself died of tuberculosis in 1849 [19,20].

No available literature mentions nontuberculous mycobacteria (NTM) and speleotherapy although NTM are emerging causative agent of mycobacterioses in humans and animals. Dust with bacteria and other microorganisms is commonly accumulated in spider’s webs, which could be a significant source of humans’ and animals’ pathogens including mycobacteria [21,22,23,24,25,26]. Current research demonstrated NTM in different matrices also in karst caves [27,28,29,30]. NTM are according to clinical relevance divided in the two risk groups: Risk Group 1 is represented by environmental saprophytes (biological agents that are unlikely to cause human diseases) and Risk Group 2 consists of potential pathogens (can cause human disease but it is unlikely to be spread throughout the community and there is usually effective prophylaxis or treatment available) [31,32].

In the Czech Republic NTM species belonging to both these Risk Groups were isolated from dust and spiders’ webs in different environments. *Mycobacterium avium* ssp. *hominissuis* was isolated from dust in house of an adult non-HIV/AIDS patient [33], *M. intracellulare* and *M.* sp. were isolated from spider webs inside of house of an immunocompromised child and from a Student Club, respectively [34]. NTM were also detected in dust and spider webs in animal houses, aviaries in zoological gardens, in henneries with domestic hens and other animal houses, stables and cages. This dust was closely associated with bedding, feeding, and other dry matrices in this environment. Contaminated dust particles with mycobacteria can be aerosolised and are important vehicle for their dissemination [23].

Cobwebs make for a relatively dry environment, which is not favourable for survival of mycobacteria [23]. It has been previously speculated that the dry environment of the spider web exacerbated by daylight and dust results in unviable conditions for mycobacterial survival [35]. This correlates with the fact that detection of acid-fast bacteria (AFB) after Ziehl-Neelsen (ZN) staining and/or the detection of DNA of different mycobacterial species often occurs without successful isolation of mycobacteria [34,36]. DNA of *M. avium* ssp. *hominissuis* and *M. avium* ssp. *M. avium* was detected by ssp. specific qPCR from cobwebs above nests in coop with old hens in different henneries [36], *M. avium* ssp. *M. avium* was detected in spider webs in neighbour’s chicken coop of one immunocompromised child [34].

Prevalence of mycobacterioses in the historical territory of Moravia and Silesia was influenced by the polluted air in this industrial part of the Czech Republic. High concentrations of aerosol particles (PM_1_, PM_2.5_, and PM_10_) in particular, are important factor for the prevalence of mycobacteriosis. PM_2.5–10_ of this size can penetrate upper lung airways and the smaller PM_1_ can penetrate the lung alveoli. Here these particulates can deposit NTM, resulting in an increase the risk for pulmonary mycobacteriosis and further, they can lead to more serious pulmonary pathological processes [37,38].

According to literature data and our knowledge the presence of mycobacteria in spider webs and dust particles in karst caves and in caves used for speleotherapy has not been investigated until now. Thus, the aim of this study was the detection, quantification and species identification of mycobacteria present in dust in spider webs and aerosol collected from different caves in the Czech Republic as well as from *Sloupsko-Sosuvsky* Cave System used for speleotherapy to assess the risk of mycobacteriosis in this scenario.

## 2. Materials and Methods

A total of 152 samples (96 spider webs and 56 aerosol/dust particulate samples) were collected. Spider’s webs originated from the surface environment from four karst systems located in the Czech Republic (*n* = 44), from photic zones of caves from the Czech Republic, Italy, and Slovakia (*n* = 26) and from aphotic zones of the caves (*n* = 26) from the Czech Republic, France, and Slovakia. A total of 87 (90.6%) of these spider’s webs samples originated from the Czech Republic. Aerosol particulate samples (*n* = 56) were sampled from filtrated air in the Czech Republic in the *Sloupsko-Sosuvsky* Cave System (speleotherapy is undertaken in this cave system; *n* = 21) and from the air collected from the large city *Brno* close by (*n* = 35).

### 2.1. Sample Collection

Spider webs were taken by sterile tongue depressor and put into sterile plastic bag. After collection, the samples were kept in a dark room at +6 °C up to one week before analysis [35].

Aerosol particles in size fraction PM_10_ and PM_2.5_ were collected as follows: From the *Sloupsko-Sosuvsky* Cave System (cave adapted for speleotherapy) 10 PM_2.5_ and 11 PM_10_ samples were collected. Aerosols were sampled in parallel with two medium volume sequential samplers (type SEQ 47/50, Sven Leckel Ingenieurbuero GmbH, Germany, flow rate of 2.3 m^3^/h) for 24 h on teflon (47 mm, 1 µm porosity, Zefluor, PALL, Ann Arbor, MI, USA) and nitrocellulose (47 mm, 1.2 µm porosity, Merck Millipore Ltd., Cork, Ireland) filters as described previously [39,40].

A total of 30 samples of PM_10_ aerosols were collected from the car park of the Transport Research Centre in *Brno* continuously for 10 days in parallel with 3 different kinds of filters, i.e., nitrocellulose (47 mm, 1.2 µm porosity, Merck Millipore Ltd., Cork, Ireland), teflon (TF 1000, 1 µm, 47 mm, PALL, Ann Arbor, USA) and quartz fiber (47 mm, Whatman, Maidstone, UK) filters, respectively, using 3 low-volume aerosol samplers (flow rate of 2.3 m^3^/h, LVS-3, Sven-Leckel Ingenieurbuero GmbH, Berlin, Germany).

Five additional samples of PM_10_ aerosols were continuously collected into deionized water using a home-made Condensation Growth Unit—Aerosol Counterflow Two-Jets Unit (CGU-ACTJU) sampler [40] on the balcony of the Institute of Analytical Chemistry in *Brno*. Each aerosol sample was collected for 9 min from 90 L of air into 9 mL of deionized water. Collected aerosols were concentrated and analysed by PCR for the presence of mycobacterial DNA.

### 2.2. Bacteriological Methods for Mycobacteria Isolation and Identification

All 152 samples were examined for the presence of mycobacteria by direct microscopy after Ziehl-Neelsen (ZN) staining, culture examination, and by mycobacterial DNA quantification by qPCR (Table 1 and Table 2).

#### 2.2.1. ZN Staining

Ziehl-Neelsen staining (ZN) was used before light microscopy for detection of acid-fast rods (AFB); at least 200 fields of view were examined in each sample [41]. The amount of AFB was evaluated as follows: negative (No AFB) and + (presence of AFB).

#### 2.2.2. Culture Examination

Spider webs samples were transferred into a 30 mL container (suitable for centrifugation), and 5 mL of distilled water was added. After vigorous vortexing for 1 min, samples were centrifuged at 500 revolutions per minute (rpm) for 10 min and supernatant was retransferred to a new container. Teflon (47 mm, 1 µm porosity, Zefluor, PALL, Ann Arbor, USA) and nitrocellulose (47 mm, 1.2 µm porosity, Merck Millipore Ltd., Cork, Ireland) filters were in whole washed by vortexing in 30 mL container in 10 mL of distilled water with approx. 20 glass bead, 4 mm in diameter. After washing filters were discarded and eluates were left in original container. Decontamination was undertaken by adding 5 mL (web eluates) or 10 mL (filter eluates) of 4% NaOH and shaking for 10 min. After centrifugation and neutralization, samples were inoculated onto four Löwenstein-Jensen media slants culture media, which were incubated at 30 and 37 °C for 12 weeks. [24]. In each sample culture positivity was evaluated according to the numbers of colony forming units (CFU’s). Different types of visible single CFU in one sample were individually subcultured for further species identification.

#### 2.2.3. Isolates Identification

The isolates were identified by molecular biological methods. Genotype *Mycobacterium* CM, AS, NTM/DR kits (Qiagene, Hain Lifescience, Nehren, Germany) were used for basic identification. Detailed identification of mycobacterial sp. and ssp. not included in above mentioned commercial kits was performed by sequencing the DNA segment encoding 16S RNA and BLAST analysis [30]. *M. avium* ssp. *avium* was further identified by the PCR method for the detection of the specific IS*901* amplicon and *M. avium* ssp. *hominissuis* for and the specific IS*1245* amplicon [42].

### 2.3. Molecular Methods for Mycobacteria Quantification

DNA isolation was performed from a 1 mL of homogenized and decontaminated eluate (described in Section 2.2.2). Isolation buffer from Anyplex mycobacterium MTB/NTM detection kit (Seegene, Seoul, Korea) was added and the suspension was heated for 20 min in a tempered heating block before this sample was used as template DNA. A total of 0.5 µL of DNA isolation product was added to a Precision PLUS 2× qPCR Master Mix (Mycobacterium spp. advanced KIT; Primerdesign Ltd., Camberley, UK), and the qPCR reaction was run with a CFX96 real-time PCR detection system (Bio-Rad Laboratories, Hercules, CA, USA) using the following thermocycler conditions: enzyme activation at 95 °C for 2 min, 50 cycles at 95 °C for 10 s for denaturation and at 60 °C for 60 s for aneling and data reading. Relative quantity expressed in copies of DNA (cp DNA) based on standard curve was calculated by CFX Manager software (Bio-Rad Laboratories, Hercules, CA, USA). Standard curve was obtained by measurement of a downwardly diluted internal standard included in Mycobacterium spp. advanced KIT (Primerdesign Ltd., Camberley, UK) [30].

### 2.4. Particulate Matter Analyses

Aerosols collected on teflon filters were extracted with 8 mL of deionized water under ultrasonic agitation for 20 min and extracts were analysed by ion chromatography (ICS-2100, Dionex, Sunnyvale, CA, USA) for 6 anions (i.e., fluoride, chloride, nitrite, nitrate, sulphate, and phosphate). Aerosols collected on nitrocellulose filters underwent gravimetric analysis to determine the mass concentrations of aerosol particles.

### 2.5. Sociological Research

The tool of our sociological research was a questionnaire survey. Data collection took place from August 2018 to May 2020. Questionnaires were distributed in printed form to parents of paediatric patients (up to 18 years of age) who were treated in the Children’s Hospital with Speleotherapy in *Ostrov u Macochy*, using the *Sloupsko-Sosuvsky* Cave System researched in this study. A total of 216 questionnaires completed by paediatric patients with asthma were evaluated including 20 questionnaires, which were distributed as a pilot survey (so-called pre-test). In connection with the presented study, we were especially interested in how the effects of the speleotherapy treatment manifested in specifically whether it led to a reduction in the amount of antibiotics administered subsequently. To this end the question was asked: “*How many times a year did the child have to be prescribed antibiotics by physicians before and after speleotherapy*”. The results of the questionnaire survey were evaluated on the basis of declaratory answers of parents of sick children. The intention was to record the consequences of speleotherapy for the quality of life of families and sick children, including in connection with the frequency of medications, without wanting to assess the medical relevance.

### 2.6. Statistical Analysis

Data analysis was performed using statistical software StatXact 12.0 (Cytel Inc., Waltham, MA, USA). *p*-values less than 0.05 were considered statistically significant. Detection rates of NTM were compared using Fisher’s exact test. For multiple comparisons tests Bonferroni adjustment of *p*-values were used.

## 3. Results

### 3.1. Bacteriological Examinations

#### 3.1.1. ZN Staining

AFB were detected in spider’s webs in all three groups of samples (Table 1); the statistically significantly (*p* < 0.05; Fisher’s exact test) highest detection rate of 46.2% was found in spider webs from photic zones of caves compared with 20.5% in spider webs from the surface environment of karst areas, and 15.4% of those collected from aphotic zones of caves. 5.7% of PM samples from aerosol collected from the city of *Brno* were positive for AFB by microscopic examination; AFB were not detected in PM collected in part of the cave with speleotherapy (Table 2).

**Table 1 microorganisms-09-02573-t001:** Examined spider’s webs in karst areas.

Spider’s Webs Origin	No. of Samples	ZN Microscopy ^4^		Culture		qPCR ^5^	
		+ve ^6^	%	+ve ^6^	%	+ve ^6^	%
Surface of Karst Areas ^1^	44	9	20.5	6	13.6	35	79.5
Photic Zones of Caves ^2^	26	12	46.2	5	19.2	16	61.5
Aphotic Zones of Caves ^3^	26	4	15.4	2	7.7	8	30.8
Total	96	25	26.0	13	13.5	59	61.5

Table interpretation. ^1^ Bohemian (*n* = 11), *Hranice* (*n* = 11), and Moravian (*n* = 22) Karst Systems in the Czech Republic; ^2^ Karst Caves in the Czech Republic: Bohemian (*Srbsko-Stary Propad*, *Lom na Chlumu-Srbska*, and *Skvira* Caves, *n* = 3), Hranice (*Zbrasov* Aragonite Caves, *n* = 17), Moravian (*Pekarna*, *Kostelik*, and *Jachymka* Caves, *n* = 3), in Sicily in Italy (Cave Monte *Conca*, *n* = 2), and in Slovakia (*Liscia Diera* Cave, *n* = 1); ^3^ Karst Systems in the Czech Republic: *Hranice* Karts (*Zbrasov* Aragonite Caves, *n* = 3) and Moravian Karst (*Sloupsko-Sosuvsky* Cave System, *n* = 16 and *Byci Skala* Cave System, *n* = 1), in France: *Duganelle* Cave (*n* = 1), and in Slovakia: *Bezstropa* Caves, *Kopcova Skala*, *Hradisko*, and *Janik* Caves (*n* = 5); ^4^ microscopy examination after the Ziehl-Neelsen staining was done in at least 200 fields [41]; ^5^ quantitative PCR test (qPCR values on a logarithmic scale detection between 3 to 4); ^6^ +ve = positive results.

#### 3.1.2. Culture Examination

From a total of 152 samples examined (Table 1 and Table 2) 13 (7.9%) were positive for isolation of NTM species; only spider’s web samples were culture positive (Table 2). The highest culture positivity (not statistically significantly; *p* > 0.05; Fisher’s exact test) was documented in photic zones of caves (19.2%) in comparison with culture positivity from webs found on the surface environment of karst areas (13.6%) and aphotic zones of caves (7.7%; Table 1). Culture examination was negative in all collected PMs, from both caves with speleotherapy facilities and from air collected around the city of *Brno* (Table 2). The environmental samples examined in this study (cobwebs and PM) did not pose a major problem in terms of contamination by other microflora. The decontamination procedure used by us was sufficient to eliminate the accompanying microflora. Contamination occurred rarely in the processed samples, and it was always possible to sub-cultivate uncontaminated CFU from the surface of the medium.

**Table 2 microorganisms-09-02573-t002:** Collected aerosol particles (Particulate Matter, PM) in from the speleotherapy facility in the *Sloupsko-Sosuvsky* Cave System and from air collected around the close by city of *Brno*.

Particular Matter Origin	No. of Samples	ZN Microscopy ^3^		Culture		qPCR ^4^	
		+ve ^5^	%	+ve ^5^	%	+ve ^5^	%
*Sloupsko-Sosuvske* Cave System ^1^	21	0	0	0	0	2	9.5
*Brno* ^2^	35	2	5.7	0	0	6	17.1
Total	56	2	3.6	0	0	8	14.3

Table interpretation. ^1^ *Sloupsko-Sosuvsky* Cave System (location of speleotherapy facility); ^2^ *Brno* (collected at the carpark of the Transport Research Centre and on the balcony on the first floor of the Institute of Analytical Chemistry at *Veveri* Street, *Brno*, Czech Republic); ^3^ microscopy examinations after the Ziehl-Neelsen staining; ^4^ quantitative PCR test, qPCR values on a logarithmic scale between 3 to 4 were considered positive for mycobacterial DNA detection [30]; ^5^ +ve = positive results.

#### 3.1.3. Detected Mycobacterial Species and Subspecies Composition and Clinical Relevance

All 11 identified isolates were NTM species. Three isolated NTM species and one complex (*Mycobacterium gordonae, M. kumamotonense, M. terrae,* and *M. terrae* complex; *n* = 5) are regarded as environmentally saprophytic mycobacteria from Risk Group 1 [31,32]. They were isolated only from spider web samples collected from the surface environment of karst areas and in photic zones of caves. Five isolated NTM species (*M. avium* ssp. *hominissuis*, *M. fortuitum*, *M. intracellulare*, *M. peregrinum* and *M. triplex*; *n* = 6) are potentially pathogenic mycobacteria in Risk Group 2 [31,32]. These NTM were found in spider webs from all three types of karst environment (Table 3).

### 3.2. Mycobacteria Quantification by qPCR

The presence of mycobacterial DNA was statistically significantly higher (*p* < 0.01; Fisher’s exact test) in spider webs (61.5%) in comparison with PMs samples (14.3%). Also the differences between the detection rate of mycobacterial DNA from spider webs samples collected from the surface of Karst areas as well as photic zones of caves in karst areas (Table 1) were statistically significantly higher (*p* < 0.01; Fisher’s exact test) than mycobacterial DNA detected in PMs samples (Table 2).

### 3.3. Characterization of Air and Aerosols in Caves

Temperature and relative humidity of air in caves was relatively constant, with a mean of 8.2 °C (range 8.0–8.4 °C) and 94.7% (range 93.6–96.6%), respectively. Mass concentrations of aerosols PM_2.5_ (i.e., PM with aerodynamic diameter of particles smaller than 2.5 µm) and PM_10_ (i.e., PM with aerodynamic diameter of particles smaller than 10 µm) collected from the caves was found. The average PM_2.5_ and PM_10_ mass concentration was 5.49 µg m^−3^ and 11.1 µg m^−3^ respectively. The concentration of 6 water-soluble anions was measured in all PM_2.5_ and PM_10_ aerosol samples collected in the cave. The mass of determined anions together accounted for about 2.88% and 2.18%, respectively, of total PM_2.5_ and PM_10_ mass. Average anion concentrations in both PM_2.5_ and PM_10_ aerosols are summarized in Table 4.

### 3.4. Sociological Research

The frequency of antibiotic use by the group of 216 paediatric patients during the calendar year was surveyed with the percentage of patients taking a course of antibiotics once a year was 19.9%, 4 times per year 15.3%, 5 times per year 19.4% and 6 times per year at 14.4%. The proportion of other frequencies of antibiotic administration was in the lower range (2.8–6.5%) and antibiotics were not required for 10.6% of patients (Table 5).

After speleotherapy, the need for antibiotic treatment in paediatric patients decreased in the calendar year. For the group of patients that previously took 1 course of antibiotics the year before speleotherapy treatment, the average for the following year decreased to 0.7 for the year. All patients from the groups that had required treatment with antibiotics 2, 3 or 4 times the year before speleotherapy treatment did not require any antibiotic treatment in the year after speleotherapy. The groups of patients that required courses of antibiotics 5, 6, 8 or 10 times in the year before speleotherapy treatment had a decrease in antibiotic use afterwards with an average of 1.4, 1.5, 2.5 and 1.2 respectively (Table 5).

## 4. Discussion

Spider webs have been used as environmental indicators for different biological contaminants for many years [43,44,45,46]; we have investigated the NTM prevalence in spider’s webs collected from the surface environment for this reason. Spider webs have been previously used as alternative method of measuring aerosol survival of filoviruses [47], bacteria [48,49,50,51] and mycobacteria, i.e., *M. avium* ssp. *paratuberculosis* [52] and NTM [24].

In this study 87 (90.6%) of these spider web samples originated from the Czech Republic so we have made comparisons with NTM prevalence in spider webs with previous results from this country. The NTM prevalence (13.6%) in spider webs collected on the surface of karst areas by culture examination in karst areas (Table 1) was compared with previously published results of NTM prevalence in different environments in the Czech Republic. NTM were cultured from 24 (8.8%) cobwebs and dust samples from households from patients with pulmonary mycobacteriosis [34,36], in aviaries in zoological gardens from birds infected with avian tuberculosis [53], in pig stables with mycobacteriosis [35,54,55,56] and hens flock with avian tuberculosis [57] in the Czech Republic (Table 6). This result is in the concordance with mycobacterial DNA detection by qPCR, with the highest amount of positive detection (79.5%) was found in spider webs collected on the surface of karst areas (Table 1).

The highest NTM prevalence in samples collected in photic zones of caves (19.2%; Table 1) could be due to an increased air flow in these locations. For NTM survival the air humidity and temperature are crucial. In dry conditions NTM quickly lose the ability to propagate on artificial media [23]. The climate of caves is different from a typical outdoor climate. Depending on the outside temperatures and humidity the air flows into or out of caves. At the mouth of the caves, the air flow is most intense with compared to the large spaces inside the caves. The average air temperatures in the Czech karst caves vary between 6.2 and 14.3 °C throughout the year and relative humidity varies between 94.6 and 99.5% [58]. These conditions could explain higher culture positivity in spider webs collected from the photic zones of caves (19.2%) compared to that of the surface of the Karst areas (13.6%) even though mycobacterial DNA was detected more often from spider webs from the latter (Table 1).

The lower cultivation uptake of NTM from cobwebs and PM could also be due to types of media used (egg-based Löwenstein-Jensen media). These were used for these examinations due to long-term experience with the examination of environmental samples for the presence of NTM [30,41]. Agar-based Middlebrook media has been raised to cultivate environmental samples until now. The routinely used metabolic culture system is stably set at 37 °C. However, because some mesophilic NTM species with a growth optimum below 35 °C could not be demonstrated by culture in environmental samples, this method was not used. Detection of only mycobacterial DNA by qPCR can be explained by the detection of only DNA residues of mycobacterial cells that are dead or non-viable [30].

Studies in the Czech Republic have shown that dust particles from the surface environment are often contaminated by NTM; from 94 dust samples examined, NTM was cultured in 6 (6.4%) samples [33,34,54]. In this study, a high prevalence of NTM was confirmed by qPCR with mycobacterial DNA detected in 79.5% of samples (Table 1). Consequent low culture positivity could be explained by variable environmental conditions (temperature, low relative humidity and day light). Especially UV light and dry conditions relative quickly devitalized NTM [23].

Inside the caves, in aphotic parts, the air flow is not so intensive [58], due to this, dust particles with NTM sediment quickly and the potential for coming into contact with a spider web is relatively lower. This correlates with NTM prevalence by culture examination and qPCR detection being lower, 7.7% and 30.8%, respectively, then the positive culture examinations (13.6%) and mycobacterial DNA detected by qPCR (79.5%) from spider webs on surface on karst areas and in photic zones of caves, 19.2 and 61.5% respectively (Table 1).

Analysed particulate anions, i.e., F^−^, Cl^−^, NO_2_^−^, SO_4_^2^^−^, PO_4_^3^^−^ and NO_3_^−^ (Table 4 and Figure 1) originate largely from the burning of wood and coal for residential heating [59] in the nearby village *Sloup* and in other villages in the surroundings. Another source of anions in PM is from motorised road in front of the cave. The presence of the studied anions in cave aerosols shown in Table 4 and Figure 1 indicates the movement of polluted air from the outer environment into the cave. The changes of PM_2.5_, PM_10_ and anions in aerosol concentrations are caused by resuspension of dust lying on the floor of the cave due to the movement of cave visitors. The cave is relatively dry on the floor and therefore the dust swirls easily [39]. Another surface possible source of particulate anions is aerosol from the *Sloupsky* Stream. Treated water from the *Sloup* communal wastewater treatment plant (located about 100 m from the cave engulfment of *Sloupsky* Stream) drains directly into *Sloupsko-Sosuvsky* Cave System. The cave for speleotherapy is located above this and air with this aerosol with different anions flows into it through the ground water system (unpublished observation).

Atmospheric PM received significant attention because of its negative effects on human health [60]. In addition, PM plays an important role in many environmental problems such as visibility reduction, global climate change, deterioration of air quality, and smog formation [59]. The environmental and health effects of atmospheric aerosols depend on the particle size and the chemical composition of particles. The chemical composition of aerosols is of particular importance to apportion their sources and to assess the health risk of aerosol constituents. Mass concentrations of aerosol particles collected in the Sloupsko-Sosuvsky Cave System both in size fraction PM_2.5_ and PM_10_ did not exceed during the measurement the daily value of 15 and 45 μg/m^3^, respectively (Table 4), which is the limit value set for both fractions by the World Health Organization to protect human health. Inorganic anions found in aerosol samples both PM_2.5_ and PM_10_ are not dangerous for human health but their presence in aerosols is important for source apportionment of aerosols; i.e., it indicates anthropogenic sources of aerosols collected in the speleotherapeutic part of the cave.

Recent research, however, suggests that most caves ventilate intensively (exchange air with the outside environment) and are far from isolated systems, as previously thought. It is possible to identify pollution from the external environment inside the caves, incl. carbon spherical particles produced by automobiles [61,62]. In the case of caves accessible to the public, visitors can be a source of pollution as well [63]. Particles can also be formed by natural processes inside the cave such as the dispersion of drip water to form a very fine aerosol [64]. This aerosol is characterized by a high calcium content, which according to some studies [65] may have a positive effect of speleotherapy. Increased activity of free calcium ions can have a beneficial effect in the calcium homeostasis of bronchial cells, and as a result, chronic inflammation and bronchial smooth muscle contraction can be reduced, which are key factors in bronchial asthma symptoms, resulting in an improvement in the clinical condition of the asthmatic patient.

In pediatric patients with asthma, there is an improvement in clinical and immunological parameters and a shift in lung function after speleotherapy [5,7,17]. This was evident by the questionnaire survey also (Table 5). The condition for such an improvement in health is a safe underground environment that meets the relevant components set for speleotherapy. Parents of sick children declare a reduction in medication, which is associated with an improvement in the quality of life in the family environment. Detailed analyses of questionnaires, the social and economic consequences of asthma, and the benefits of speleotherapy are therefore regularly carried out by these speleotherapy facilities [unpublished findings].

Based on previous experimental experience, NTM were detected in dust captured in spider webs on the surface of the karst environment, in the photic and aphotic parts of caves by all three methods used (microscopic, culture and qPCR examinations). In aerosol particles (PM) inside the cave, NTM were detected by qPCR in only 9.5% of samples. Therefore, it is possible to consider the cave environment in which speleotherapy was implemented from the point of view of the occurrence of NTM as safe.

## Figures and Tables

**Figure 1 microorganisms-09-02573-f001:**
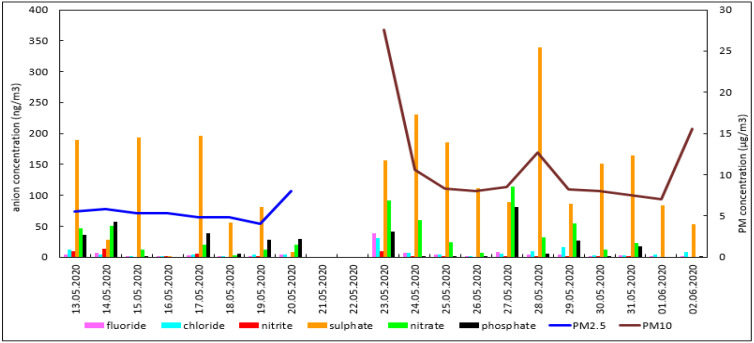
Mass concentration of PM_2.5_ and PM_10_ aerosols and concentration of anions in PM in cave with the speleotherapy facility in the *Sloupsko-Sosuvsky* Cave System. Figure interpretation. *X*-axis: Date of PM sample collection. Vertical bars correspond to the daily mean concentration of anions; Blue (Brown) line indicates the mean mass concentration of PM_2.5_ (PM_10_) aerosols.

**Table 3 microorganisms-09-02573-t003:** Mycobacterial species detection in examined spider’s webs on karst surface and in photic and aphotic zones of caves.

Species Identification	Risk Group ^1^	No. of Isolates from Spider’s Webs			
		Total (%)	Surface	Photic	Aphotic
*M. gordonae*	1	2 (14.3)	2	0	0
*M. kumamotonense*	1	1 (7.1)	0	1	0
*M. terrae*	1	1 (7.1)	1	0	0
*M. terrae* complex	1	1 (7.1)	0	1	0
*M. avium* ssp. *hominissuis* ^2^	2	1 (7.1)	1	0	0
*M. fortuitum*	2	2 (14.3)	0	2	0
*M. intracellulare*	2	1 (7.1)	1	0	0
*M. peregrinum*	2	1 (7.1)	0	0	1
*M. triplex*	2	1 (7.1)	1	0	0
*M.* sp.	NK	3 (21.4)	1	1	1
**Total (%)**		14 (100)	7 (50.0)	5 (35.7)	2 (14.3)
**Risk Group 1**	4 sp.	5 (35.7)	3	2	0
**Risk Group 2**	5 sp.	6 (42.9)	3	2	1
**Risk Group NK**	NK sp.	3 (21.4)	1	1	1

Table interpretation. *M. = Mycobacterium*; ssp. = subspecies; sp. = species; NK = Not Known *Mycobacterium* sp.; ^1^ according to [31]; ^2^ identified according to Slana et al. [42].

**Table 4 microorganisms-09-02573-t004:** Mass concentration of PM_2.5_ and PM_10_ aerosols and concentration of anions in PM in *Sloupsko-Sosuvsky* Cave System.

Sample No.	PM Size	PM Concentration	Anion Concentration in PM						qPCR
			(ng/m^3^)						cp/mL
		(µg/m^3^)	Fluoride	Chloride	Nitrite	Sulphate	Nitrate	Phosphate	DNA
1	2.5	5.6	4.16	11.8	10.1	190	46.1	36.1	3
2	2.5	5.9	7.29	4.60	13.4	27.9	50.5	58.0	0
3	2.5	5.3	1.62	1.28	<LOD	194	12.7	<LOD	0
4	2.5	5.4	0.64	1.56	<LOD	0.82	<LOD	<LOD	2
5	2.5	4.8	3.69	3.90	5.67	197	20.9	38.8	0
6	2.5	4.8	1.11	2.01	<LOD	56.4	2.69	5.11	2
7	2.5	4.1	2.04	3.91	0.36	81.8	12.3	27.7	0
8	2.5	8.0	4.42	3.96	<LOD	8.01	20.8	29.8	0
9	10	27.6	38.6	30.5	9.48	156	92.0	41.9	0
10	10	10.6	7.00	7.37	0.25	230	60.2	2.44	0
11	10	8.3	4.39	4.14	2.28	185	24.5	1.16	2
12	10	8.0	1.14	1.73	<LOD	111	7.15	<LOD	0
13	10	8.5	8.42	5.97	0.45	89.0	115	81.0	0
14	10	12.7	3.95	10.1	1.80	339	32.3	5.23	2
15	10	8.3	4.88	16.3	0.46	86.0	54.8	27.5	0
16	10	8.0	2.27	3.08	0.26	151	12.8	0.78	0
17	10	7.6	3.16	3.06	0.51	164	22.8	18.0	3
18	10	7.0	1.56	4.88	<LOD	83.4	<LOD	<LOD	0
19	10	15.6	1.63	8.59	<LOD	53.9	<LOD	0.90	0

Table interpretation. qPCR = quantitative PCR test; cp/mL DNA = number of copies per mL of mycobacterial DNA; LOD = limit od detection; qPCR values are on a logarithmic scale. LOD of nitrite, nitrate and phosphate is 0.19, 0.17 and 0.26 ng/m^3^, respectively.

**Table 5 microorganisms-09-02573-t005:** Evaluation of survey of 216 paediatric patients with asthma treated by speleotherapy.

Repetitions of Antibiotics’ Administrations	No. of Patients	Antibiotics’ Administrations ^1^	
		Before	After
		Speleotherapy (%)	Speleotherapy (Average)
0	23	10.6	0.0
1	43	19.9	0.7
2	14	6.5	0.0
3	12	5.6	0.0
4	33	15.3	0.0
5	42	19.4	1.4
6	31	14.4	1.5
7	0	0	0
8	12	5.6	2.5
9	0	0	0
10	6	2.8	1.2
Total	216	100	

Table interpretation. ^1^ Antibiotics’ administration one year before and one year after the speleotherapy treatment in the Children’s Hospital with Speleotherapy in *Ostrov u Macochy*, Czech Republic.

**Table 6 microorganisms-09-02573-t006:** Mycobacterial species detection in examined spider’s webs on karst surface and in photic and aphotic zones of caves.

Examined Vehicles ^1^	Cultured Samples		Mycobacterial Species (No.)	Reference
	No.	+ve ^2^		
Cobwebs	4	1	M. sp. (1)	[36]
Spider nets	13	2	MIN (1), M. sp. (1)	[34]
Web and dust	23	1	MAH ^3^ (1)	[53]
Dust and spider webs	117	9	MAH ^3^ (3), MFO (2),	[35]
Dust and spider webs	17	2	M. sp. (2)	[55]
Dust and spider webs	88	9	MAH ^3^ (4), MTR (1), MNO (1), MTE (1) M. sp. (2)	[56]
Spider nets	9	0		[54]
Spider nets	1	0		[35]
Total	272	24		
%	100	8.8		

Table interpretation. MAH *=*
*M. avium* ssp. *hominissuis*; M. = *Mycobacterium*; sp. = species; MTR = *M. triviale*; MNO = *M. nonchromogenicum*; MTE = *M. terrae*; MFO = *M. fortuitum*; MIN = *M. intracellulare*; ^1^ as vehicles are described in published source; ^2^ +ve = positive results; ^3^ identified according to Slana et al. [42].

## Data Availability

Availability of data and materials correspondence and requests for mycobacterial isolates be addressed to the corresponding author.

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
