# Peer review of "Nontuberculous Mycobacteria Prevalence in Aerosol and Spiders’ Webs in Karst Caves: Low Risk for Speleotherapy"

_microorganisms, 2021, doi:10.3390/microorganisms9122573_

Round 1

Reviewer 1 Report

Overall I found it very hard to follow the point of this manuscript. The introduction is far too long and contains too much superfluous or irrelevant information. It took a long time to get to the aims of the study, and much of the background on the benefits of speleotherapy have nothing to do with the detection of NTM. 

Suggest you focus on

  • What is speleotherapy?
  • How is it used?
  • What is the significance of the spiders webs and why would you sample them?
  • NTM cause disease, and does their presence in the caves pose a risk for the patients who use the caves

Methods

See comments on annotated manuscript regarding the limitations of culture, particularly using only one (LJ) media- the yield for culture of NTM from environmental samples could have been enhanced by using 7H11 agar and liquid (MGIT) culture.

Bacterial and fungal overgrowth is a common problem yet the proportion of samples affected is not stated.

The sociological aspect to the paper is very weak. The question asked is very leading, there is no objective assessment of antibiotic use- e.g physician prescribing records, pharmacy dispensing records etc. The frequency of antibiotic use also does not have anything to do with the risk of acquiring NTM infection from the caves, which would seem to be the point of sampling the caves for NTM.  The aim stated at the end of the introduction reads- ‘to detect, quantify and identify mycobacteria in webs and aerosols from caves and assess the risk of mycobacteriosis’ – this risk has  not been addressed in the manuscript.

The conclusion section needs to refer back to the aims of the study, at present it refers to other work, and results, but is not a discussion of the significance, if any, of the results presented. 

Author Response

We thank for all comments, which were included in the revised manuscript.

Reviewer 2 Report

The paper entitled “Nontuberculous Mycobacteria Prevalence in Aerosol and Spiders’ webs…..” by Hubelova et al. describes a survey carried out on karst caves about the prevalence in these areas of nontuberculous mycobacteria. Despite I have found the paper interesting, there are some issues I would suggest should be considered before publication:

Row 212: it is not clear how the quantification of the DNAs has been carried out;

Row 226: has the questionnaire been validated before its sub ministration?

Row  362: the prevalence here was defined by PCR?

Row 388-400: I believe here a further comment about the PM and Anions concentrations considered dangerous for human beings could be beneficial.

Author Response

We thank for all comments, which were included in revised manuscript.
